# Dissecting ribosomal particles throughout the kingdoms of life using advanced hybrid mass spectrometry methods

Michiel van de Waterbeemd[1,2], Sem Tamara[1,2], Kyle L. Fort[1,2,3], Eugen Damoc[3], Vojtech Franc[1,2], Philipp Bieri[4], Martin Itten[4], Alexander Makarov[1,3], Nenad Ban [4] & Albert J.R. Heck [1,2]

Biomolecular mass spectrometry has matured strongly over the past decades and has now reached a stage where it can provide deep insights into the structure and composition of large cellular assemblies. Here, we describe a three-tiered hybrid mass spectrometry approach that enables the dissection of macromolecular complexes in order to complement structural studies. To demonstrate the capabilities of the approach, we investigate ribosomes, large ribonucleoprotein particles consisting of a multitude of protein and RNA subunits. We identify sites of sequence processing, protein post-translational modifications, and the assembly and stoichiometry of individual ribosomal proteins in four distinct ribosomal particles of bacterial, plant and human origin. Amongst others, we report extensive cysteine methylation in the zinc finger domain of the human S27 protein, the heptameric stoichiometry of the chloroplastic stalk complex, the heterogeneous composition of human 40S ribosomal subunits and their association to the CrPV, and HCV internal ribosome entry site RNAs.

[1] Biomolecular Mass Spectrometry and Proteomics, Bijvoet Center for Biomolecular Research and Utrecht Institute for Pharmaceutical Sciences, Utrecht University, Utrecht 3584CH, The Netherlands. [2] Netherlands Proteomics Center, 3584CH Utrecht, The Netherlands. [3] Thermo Fisher Scientific, 28199 Bremen, Germany. [4] Department of Biology, Institute of Molecular Biology and Biophysics, ETH Zurich, 8093 Zurich, Switzerland. These authors contributed equally: Michiel van de Waterbeemd, Sem Tamara. Correspondence and requests for materials should be addressed to A.J.R.H. (email: a.j.r.heck@uu.nl)

**B**iomolecular mass spectrometry (MS) has matured substantially over the past decades finding applications in biochemistry, molecular and structural biology, and systems biology[1]. With its ability to analyze biological systems at multiple levels—whether its metabolites, RNA and DNA, proteins, protein complexes or entire proteomes—the MS toolbox has proven invaluable in a life science research environment[2–4]. While methods like (phospho)-proteomics and metabolomics have firmly settled in the field of cell biology, the use of MS methods for characterizing protein complexes in a structural biology setting is less matured. With recent developments in data analysis for cross-linking MS, this technique is rapidly gaining popularity among structural and systems biologists for its ability to map protein–protein interactions on a global scale[5–7]. However, other MS approaches can also complement structural biology techniques and provide highly useful insight into the assembly and composition of macromolecular assemblies[8]. Here, we describe a three-tiered MS approach for the detailed characterization of protein complexes and highlight its use by characterizing various ribosomal particles from different organisms and organelles.

Ribosomes are large ribonucleoprotein complexes responsible for the translation of messenger RNA (mRNA) into proteins. Their composition and architecture vary along the phylogenetic tree, from eukaryotes to bacteria as well as among the organellar ribosomes, although their functional elements catalyzing the key reactions like the decoding of the mRNA and the formation of the peptide bond are highly conserved[9,10]. Recent developments in structural biology techniques, notably X-ray crystallography and cryo-electron microscopy (cryo-EM), have provided insight into the structure and function of many ribosomal complexes and in combination with biophysical and biochemical data led to a detailed understanding of the translation mechanism[11]. Yet, even with structures of ribosomes from many kingdoms of life and different organelles resolved[12–15], small but potentially important features of ribosomal particles have been mostly overlooked. These features, including specific post-translational modifications (PTMs), sequence variations, binding of protein cofactors or sub-stoichiometric presence of ribosomal proteins, can be elusive to standard structural biology techniques and therefore require the use of complementary approaches, such as mass spectrometry (MS).

Our three-tiered MS approach makes use of a set of MS techniques, which provide information on the composition, assembly, and activity of ribosomal particles (Fig. 1). First, bottom-up liquid chromatography-tandem mass spectrometry (LC-MS/MS), a MS technique commonly used in proteomics research, provides the ability to identify and quantify the ribosomal proteins and their PTMs[16]. Moreover, it can determine the presence of ribosome-interacting factors, which have remained bound to the ribosomal

particles during their purification[17]. Common bottom-up LC-MS/MS workflows start with unfolding of the proteins followed by their digestion into peptides. These peptide mixtures are separated using high-performance liquid chromatography (HPLC) and sequenced by a mass spectrometer. For the second tier, top-down LC-MS/MS, proteins are denatured but kept intact and separated by a HPLC system. The intact masses of the different proteins and their co-occurring proteoforms are measured by the mass spectrometer, providing an overview of all the different versions of the gene products such as proteins carrying multiple PTMs. Proteins are identified in top-down LC-MS/MS through top-down sequencing, which can additionally localize PTMs[18–20]. In this way, top-down LC-MS/MS can potentially provide information on proteoforms of ribosomal proteins that would have been lost upon digestion into peptides like the crosstalk between different PTMs[20]. The third tier in the approach, native MS, omits even the unfolding step and introduces the intact ribonucleoprotein complexes into the mass spectrometer, after which their masses are measured[21,22]. Because non-covalent interactions are generally preserved in native MS, accurate mass measurements of the complexes can provide detailed insight into their composition, including the stoichiometry of the protein or nucleic acid subunits[23–25]. Moreover, using the latest innovations in mass-analyzers, native MS can resolve and characterize co-occurring assemblies, such as ribosomes with and without an interacting protein or with substoichiometric presence of a ribosomal protein, as was recently demonstrated for prokaryotic ribosomal particles[26].

Here, we demonstrate how this three-tiered MS approach can provide in-depth characterization of four distinct ribosomal particles: *E. coli* cytosolic 70S ribosomes (Ec70S), chloroplastic 70S ribosomes from spinach (So70S), and human cytosolic 40S (Hs40S) and 60S (Hs60S) ribosomal subunits (Table 1).

## Results

Before describing the in-depth analysis of the ribosomal particles by our three-tiered MS approach, we first describe some novel workflows, hardware and software we used for top-down LC-MS/MS making use of a recently introduced mass analyzer, the Orbitrap™HF-X[27].

**Improved top-down LC-MS/MS through on-the-fly deconvolution.** Standard top-down LC-MS/MS experiments encounter a number of challenges that limit the analytical depth of the analyses. When full MS spectra are acquired to determine which proteoforms are selected for subsequent top-down sequencing events, there is an information redundancy that comes from the repeated selection and fragmentation of the different charge states of the same proteoform. Additionally, current workflows regularly employ full MS scans at

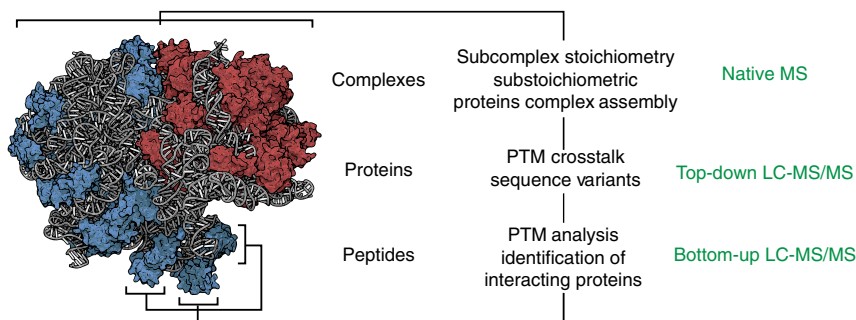

**Fig. 1** Three-tiered mass spectrometry based approach for the dissection and characterization of ribosomal particles. The three-tiered hybrid mass spectrometry approach described here can provide insight into multiple molecular levels, ranging from the amino acid sequence of individual ribosomal proteins to the stoichiometric composition of the intact ribonucleoprotein particles. Which tier in the approach provides information on which level is indicated in green. The structural model of the *E. coli* 70S ribosome (PDB 4YBB) displays the rRNA in gray, the proteins of the 30S subunit in red and of the 50S subunit in blue[70]

| Table 1 Information on the investigated ribosomal particles | | | | |
|---|---|---|---|---|
| | **Ec70S** | **So70S** | **Hs40S** | **Hs60S** |
| Organism | *E. coli* | *S. oleracea* | *H. sapiens* | *H. sapiens* |
| Subcellular location | Cytosol | Chloroplast | Cytosol | Cytosol |
| Number of proteins | 55 | 57 | 33 | 47 |
| rRNA fragments | 23S, 5S, 16S | 16S, 23S, 4.5S, 5S | 18S | 28S, 5S, 5.8S |
| Approximate mass, kDa | 2302 | 2435 | 1212 | 2688 |

high resolving power ($R = 120{,}000$ at $m/z = 200$) to determine the proteoform charge state from the isotopic distributions, followed by top-down sequencing also performed at high resolving power (so-called "High-High" workflows). However, on Orbitrap™ mass spectrometers, the short-lived transients of large proteins (>30 kDa) are generally suppressed when competing with the longer lived transients of smaller proteins, introducing a negative bias in the identification of larger proteins. Additionally, to isotopically resolve the high molecular weight proteins (>60 kDa), longer transient times are required (>500 ms) that are generally less compatible with the LC timescales of top-down LC-MS/MS analysis. By acquiring the full MS spectra at medium resolution ($R = 7500$ at $m/z = 200$), but the MS/MS data at high resolving power (termed "Medium-High" workflows), the bias is removed and both large and small proteins can be identified by top-down sequencing. At this resolution, the isotopic distributions of the proteoforms remain unresolved. This information is used to calculate their charge states, an essential part of the experiment since this information is required for the determination of fragmentation parameters (e.g., collision voltage), database searching (calculation of the measured mass) and the sequencing of co-eluting proteins (exclusion of different charge states of the same proteoform). We solved these redundancy and bias issues by actively (on-the-fly) deconvoluting the medium-resolution full MS spectra, assigning charge and mass to every peak in the spectrum (Methods section).

We benchmarked the combination of these Medium-High and High-High workflows by identifying all 55 ribosomal proteins from the Ec70S ribosome and an additional 12 ribosome-associated proteins (Supplementary Fig. 1). The advantage of the Medium-High workflow over the High-High workflow for high molecular weight proteins is immediately evident when medium- and high-resolution full MS scans of the ribosomal protein S1 (61 kDa) and the peptide chain release factor 2 (prfB, 41 kDa), a ribosome-interacting protein, are compared (Supplementary Fig. 2). While the medium-resolution scans contain clearly resolved charge states, the high-resolution scans suffer from poor signal-to-noise ratio hampering identification. Moreover, the Medium-High method allows for shorter duty cycles (Supplementary Fig. 3a) and faster processing time of the deconvolution algorithm (Supplementary Fig. 3b). On the other hand, the high-resolution full MS of High-High workflows provides a higher number of reliably deconvoluted proteoforms for lower molecular weight proteins (Supplementary Fig. 3c) making the two approaches somewhat complementary (Supplementary Fig. 4). Therefore, we used a combination of optimized Medium-High and High-High workflows to perform in-depth top-down LC-MS/MS analysis of the So70S, Hs40S and Hs60S ribosomal particles.

**Overview of bottom-up and top-down LC-MS/MS on ribosomes**. In Fig. 2, we provide an overview of the results of the bottom-up (2a–c) and top-down (2d–f) LC-MS/MS experiments on So70S, Hs40S, and Hs60S ribosome samples. On the left, the bottom-up LC-MS/MS data are presented by intensity-based absolute quantification (iBAQ) plots, which rank the detected proteins by their estimated abundance[28]. Although these plots are not directly suitable for the determination of protein stoichiometry in the complexes, they provide an accurate prediction of the protein abundances in the investigated samples. On the right, the base peak intensity chromatograms of the top-down LC-MS/MS runs are shown, where peaks represent proteins or mixtures of chemically similar proteins eluting from the column and introduced into the mass spectrometer, in which Medium-High or High-High workflows are used to measure their intact mass and perform top-down sequencing. In this way, we could identify all expected 57 ribosomal proteins of the So70S ribosome, 47 ribosomal proteins of the Hs60S and 33 ribosomal proteins of the Hs40S ribosomal subunit. Moreover, we detected several non-ribosomal proteins that co-purified with the ribosomal particles, e.g., translation factor pY (pY) and ribosome recycling factor (RRF) in the So70S sample. Such a high identification rate lays the foundation for an in-depth investigation of the co-occurring proteoforms in these ribosomal assemblies.

Information on the purity of the sample can be extracted from both bottom-up and top-down LC-MS/MS measurements. According to bottom-up LC-MS/MS data, the 50 most abundant proteins in the So70S sample are either ribosomal proteins of the 30S and 50S subunit, which comprise the 70S ribosome, or translation factors (pY and RRF). The main impurities seem to be non-ribosomal, generally high abundant chloroplastic proteins (Fig. 2a and Supplementary Data 1). Additionally, a small set of low abundant proteins of the cytosolic 80S ribosome are detected. The majority of the proteins in the Hs40S sample belong to the 40S ribosomal subunit, followed by a number of mitoribosomal proteins of the 39S large subunit (Fig. 2b). The Hs60S sample seems to be most pure since there is a much larger gap between the abundance of the 60S ribosomal proteins and other co-purified proteins (Fig. 2c). Additionally, the main impurities here include ribosomal proteins of the 40S subunit and not mitoribosomal proteins. The top-down LC-MS/MS runs show highly comparable results when inspecting the identified proteins. In the human 40S sample, around half of the ribosomal proteins identified were mitochondrial. For the 60S sample, only 12% was mitochondrial while 32% was identified as 40S ribosomal protein.

In this way, other, non-ribosomal protein complexes can also be identified. For instance, top-down LC-MS/MS showed that the purified human 40S ribosome sample contained multiple protein subunits from the spliceosome (snRNPs E, F, G, SM D1, and SM D2). Characterization of these proteins is not limited to identification; the snRNP SM D1 was found to be dimethylated nine times in its glycine- and arginine-rich C-terminus and the fragmentation maps of the snRNP E protein suggest that its initiator methionine is removed and the protein is both acetylated and dimethylated (Supplementary Fig. 5). The combination of top-down and bottom-up LC-MS/MS allows for better coverage of the proteome of these complexes. Bottom-up LC-MS/MS provides great depth to characterize lower abundant proteins and has no upper limit in protein

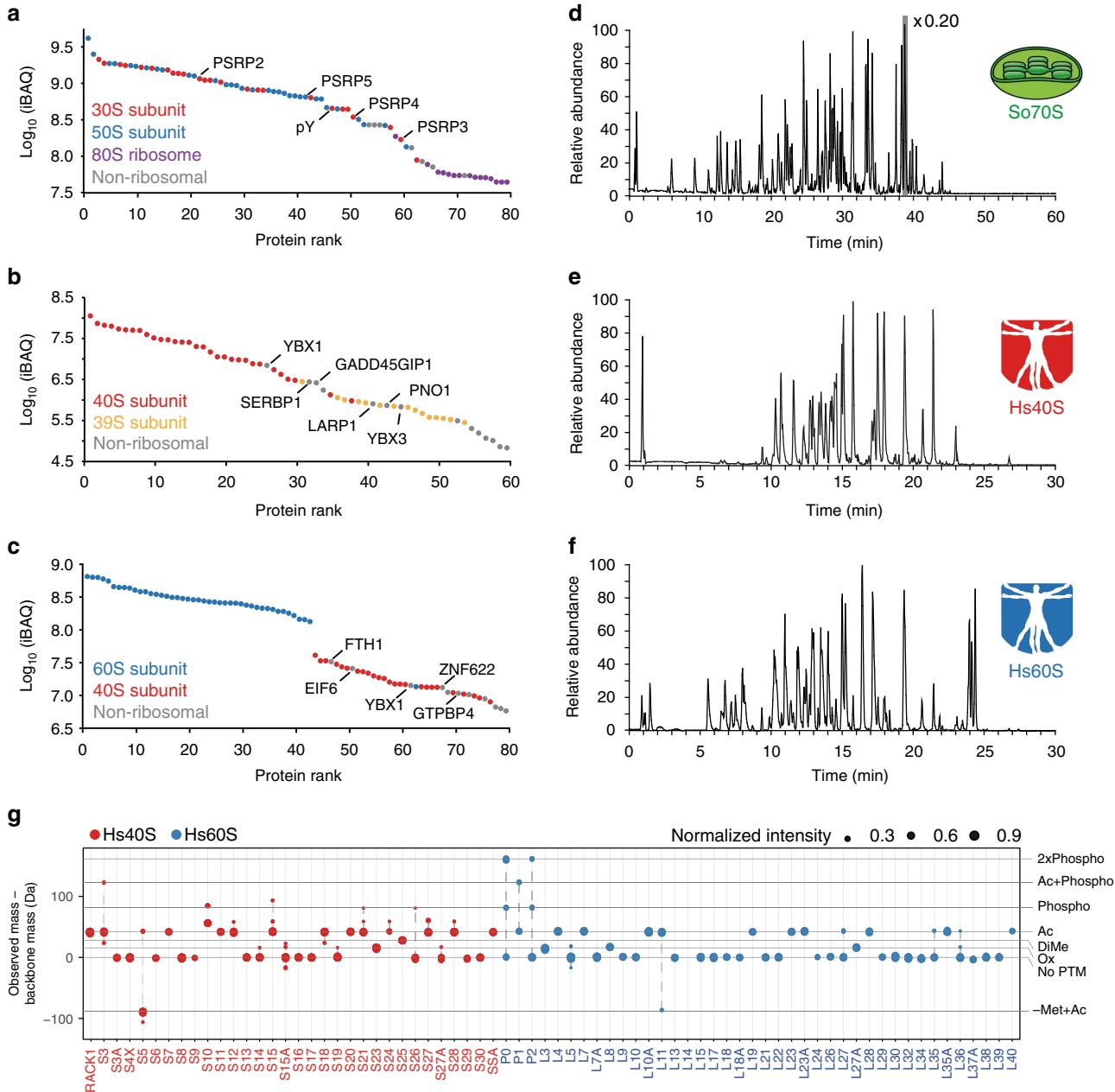

**Fig. 2** Dissecting ribosomal particles from different organisms and organelles by bottom-up and top-down LC-MS/MS analyses. **a–c** Relative quantification of both ribosomal and non-ribosomal proteins present in the preparations of So70S, Hs40S, and Hs60S ribosomal particles, respectively. Protein abundance was estimated by using the intensity-based absolute quantification (iBAQ) values of each identified protein. **d–f** Top-down LC-MS/MS base peak chromatograms of all proteins in the So70S, Hs40S, and Hs60S ribosomal particles, respectively. Top-down LC-MS/MS analysis allows for identification of both ribosomal and non-ribosomal proteins and their distinct proteoforms. **g** Relative abundance distributions of proteoforms detected in ribosomal proteins of the 40S and 60S subunit identified by database searching. Each dot represents a proteoform of the gene products listed on the x-axis with increasing size representing increasing relative abundance. The position of the dots along the y-axis shows the deviation of measured mass of the proteoform from the mass calculated from the amino acid sequence. Several commonly occurring post-translational modifications (e.g., acetylation, phosphorylation) are annotated with their corresponding mass shift. The data point for L14 is missing since the observed mass shifts introduced by a varying number of inserted alanine repeats is outside the scale of the plot (namely: +213 and +355 Da). To stay consistent, we adopted the ribosomal protein names from Uniprot entries. The pictograms of the Vitruvian man were prepared based on an image that was obtained under a CC0 license from https://commons.wikimedia.org/wiki/File:Digisapiens.png

length and molecular weight, while top-down LC-MS/MS provides a more complete view of the different co-occurring proteoforms of proteins, albeit working still best within ~5–50 kDa mass range (Supplementary Fig. 6).

An overview of proteoforms detected and characterized with top-down LC-MS/MS of ribosomal proteins in the Hs40S and Hs60S

subunits is displayed in Fig. 2g. For only half of the proteins, the intact measured mass agreed with the mass determined directly from the gene sequence. We could identify several PTMs and sequence variations, which are poorly described in most protein databases. These include the removal of initiator methionines (L11, L19, L23, L35a, L30, L36a, S5, and S25) as well as N-terminal acetylation

(L23, L35a). Absence of this information from the protein database can in turn lead to incorrect interpretation by standard database searching software, as is exemplified by the S25 protein (incorrect assignment of an N-terminally acetylated proline prevented the identification of a previously unreported dimethylated state of this S25 protein (Supplementary Fig. 7). Furthermore, available protein databases such as UniProt and neXtProt mostly lack detailed information on disulfide bridges in human ribosomal proteins. In our LC-MS/MS approach we could chromatographically separate oxidized and reduced forms of human ribosomal proteins L5, S21, S27 and S27a as well as detect their subtle mass differences (Supplementary Fig. 8).

**Determining sequences of plastid ribosomal proteins**. An advantage of bottom-up LC-MS/MS for characterizing proteins is that their initial identification can proceed by detecting just a single or a few unique peptides. Even when the exact full sequences of the proteins are not known or available, for instance because the proteome of the source organism is poorly described, a proteome from a closely related organism can be sufficient. The So70S proteins are partially encoded on the chloroplast genome while some others are encoded on the nuclear genome and have to be imported into the chloroplast. These imported proteins use a transit peptide at the N-terminus to pass the chloroplast membranes, after which this peptide is cleaved off[29]. Identification of these proteins in bottom-up LC-MS/MS is not really hampered by these events since it can be based on the part of the sequence that is not part of the transit peptide. Top-down identification however requires more accurate information because sequence variations in the termini prevent matching of fragment ions with their theoretical expected masses. On the other hand, top-down LC-MS/MS can exactly characterize the fully processed form of the ribosomal proteins, which may be beneficial in fitting the electron density maps of cryo-EM reconstructions.

Although spinach is being extensively used for structural studies, for example chloroplastic ribosomes for cryo-EM analysis are often extracted from spinach leaves[14,30], the spinach proteome is rather poorly described. At present Uniprot contains only 286 reviewed entries from *S. oleracea*, compared to 20,214 from human or 15,423 from the plant *Arabidopsis thaliana*. Although a significant part of the So70S proteins is described in the database, particularly proteins encoded in the nuclei have incomplete or missing protein sequences[31–33]. For these proteins, the Beta Vulgaris Resource (BvSeq), which contains a set of spinach genome sequencing data, is a helpful source[34]. However, this database does not contain the sequences of a subset of the So70S proteins encoded on the chloroplast genome. Additionally, gene sequences from the two different sources are not always identical and BvSeq does not distinguish between the transit peptide and the protein product sequence.

We analyzed our top-down LC-MS/MS data with Proteome Discoverer Absolute Mass Search, which features database searching with an extended precursor mass window[18]. A sufficiently large window (5 kDa) allows the identification of plastid ribosomal proteins even when N-terminal processing is not taken into account, by making use of fragments from the C-terminus. Furthermore, if processing of the N-terminal transit peptide is taken into account it can also identify processing on the C-terminal side correctly. In this way, we attempted to determine the N- and C-terminal processing of the ribosomal proteins in our So70S preparation, as well as their potential PTMs. We managed to identify all 57 ribosomal proteins of the chloroplastic 70S ribosome including five plastid-specific ribosomal proteins (PSRPs) as well as the translation factor pY (formerly PSRP1) and

the ribosome recycling factor (RRF) using top-down sequencing and assembled their protein product sequences (Fig. 3a, Supplementary Data 2). Notably, in the reported cryo-EM structure of So70S, no electron density was annotated for RRF while it is readily identified in our top-down LC-MS/MS runs[14].

Although in the majority of the cases the correct sequences of the So70S proteins could be extracted from the combination of Uniprot and BvSeq databases, some proteins required more extensive manual interpretation. Protein L27 is encoded in the nucleus and identification of the protein only occurred when the first 58 amino acids (determined previously to be the transit peptide) were removed (Fig. 3b, c). However, removal of these amino acids still left a ~1.2 kDa mass difference between the measured mass and the sequence mass and no C-terminal fragments could be detected. By removing amino acids 182–194 this mass difference could be accounted for and 10 fragment ions from the C-terminus were newly identified. Interestingly, the online transit peptide prediction tool ChloroP also predicts the presence of a cleavage site between amino acids 181 and 182, suggesting similar processing enzymes may have performed the cleavage[35]. Protein S6 is encoded in the nucleus and is reported to exist in multiple forms resulting from distinct transit peptide cleavages (Fig. 3d, e). In line with this, the protein was identified with up to 9 kDa mass differences, only C-terminal fragment ions and in five different forms with ranging abundance. Uniprot reports five different N-termini at positions 66, 71, 77, 86, and 91 while ChloroP predicts transit peptide cleavage sites between amino acids 61 and 62. By combining the intact masses of the S6 proteoforms with their corresponding fragmentation maps we could determine the N-terminal processing in this ribosomal protein. The most abundant form of S6 has Ala-61 as N-terminal amino acid, as predicted by ChloroP, and has additional N-terminal acetylation. Three additional S6 variants could be detected, which all have a single amino acid less at the N-terminus, suggesting they are likely a result of exo-proteolytic processing rather than differential transit peptide cleavages. A fifth form starts with Ser-68 and is N-terminally acetylated while the last, second most abundant form, which features Pro-83 at the N-terminus, lacks acetylation. The data on chloroplastic ribosomal protein S6 nicely illustrate the richness of the ribosomal proteome through co-occurring proteoforms and the benefit of top-down characterization.

**Novel cysteine methylations in human ribosomal proteins**. The advantage of top-down LC-MS/MS over bottom-up LC-MS/MS is not just limited to the detection of unknown sequence processing events. Ribosomal proteins can also exist in multiple proteoforms as the result of PTMs. These modifications can be detected in bottom-up MS but by digesting the protein information on any potential crosstalk between modified sites is lost. In top-down LC-MS/MS, all proteoforms with distinct masses can be detected based on their intact mass. Information on the site and occupancy of the modifications can be gathered from the fragmentation spectra. A good example of this is the human ribosomal protein S27 that we found to be multiply methylated (Fig. 4). Initially, S27 was identified with a mass around 42 Da higher than the mass predicted from the sequence, hinting at acetylation. However, fragmentation maps indicated removal of the initiator methionine leaving an N-terminal proline, making N-terminal acetylation unlikely[36]. Close inspection of the proteoforms identified as S27 revealed an additional variant with a mass difference around 28 Da, albeit lower abundant. Since this suggested methylation rather than acetylation we performed an in-depth analysis of the top-down fragmentation spectra. By comparing the masses of fragments from both the N- and the C-terminus with their

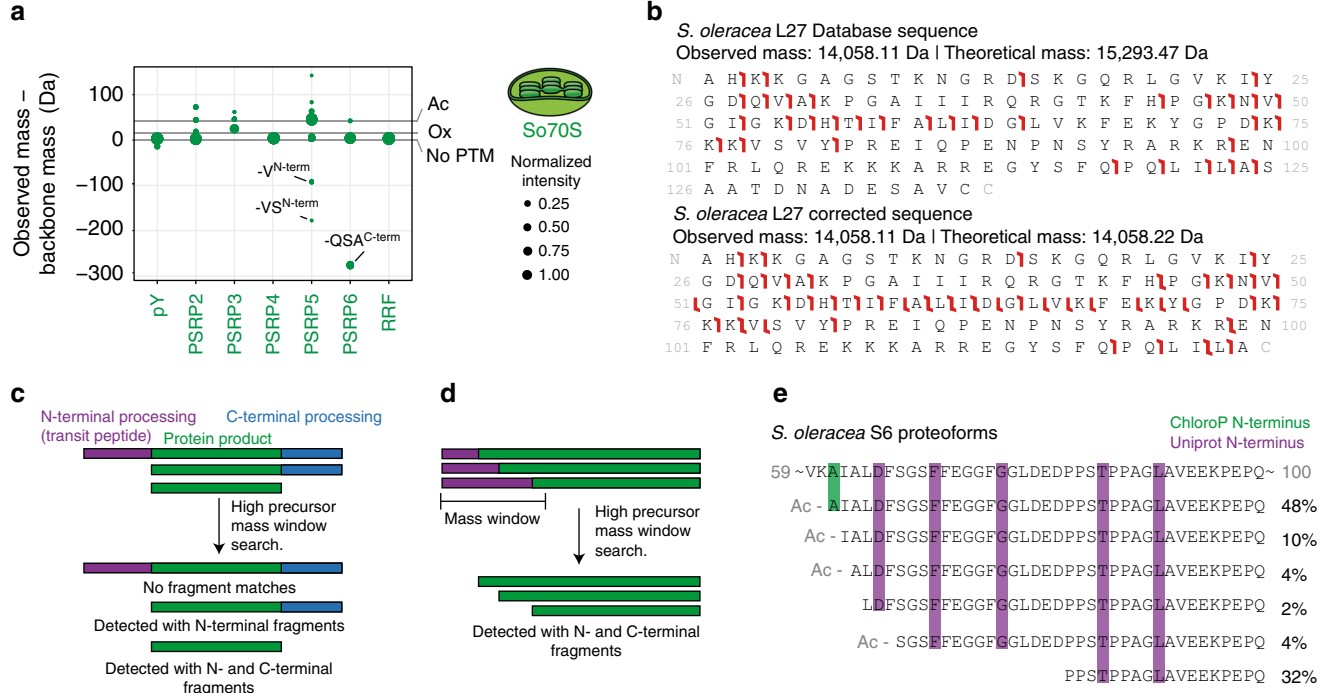

**Fig. 3** Top-down analysis using high precursor mass window searches identifies processing of the plastid ribosomal proteins. **a** For the plastid-specific ribosomal proteins (PSRPs) of the chloroplastic 70S ribosome, the translation factor pY and the ribosome recycling factor (RRF), several co-occurring proteoforms could be identified and quantified, resulting from post-translational modification and/or processing of the termini. In the corresponding cryo-EM structure of the chloroplastic 70S ribosome, RRF was not modelled due to absence of electron density[14]. **b, c** The chloroplastic ribosomal protein L27 is processed at both the N- and C-terminus and is therefore not detected through top-down sequencing when searching for the intact gene product. Corrections to either the N- or C-terminal sequence provides fragment ions from the C- and N-terminus, respectively, as long as the precursor mass window is larger than the mass of the sequence correction. **d, e** In a similar way, variable processing at the N-terminus of the chloroplastic ribosomal protein S6 could be detected by using C-terminal fragments to first identify which intact masses belong to S6 and subsequently correct the sequence until the theoretical and measured masses matched

theoretical masses based on the S27 sequence, we found fragments of the protein carrying between one and three modifications with a mass of 14 Da. We assigned these modifications as methylations to three cysteines at positions 39, 55, and 58 (Fig. 4b). Interestingly, these cysteines are all part of a C4-type zinc finger domain that protrudes from the side of the 40S subunit (Fig. 4c). Methylation likely prevents the binding of a zinc ion, which agrees well with the electron density map[37]. Although methylation of cysteine residues in S27 has been reported in yeast ribosomes, methylation was never reported for human ribosomes, nor to this extent[38]. This is likely caused by the fact that in standard bottom-up LC-MS/MS workflows cysteine methylation is not commonly included in the database search. The role of this modification is unknown although methylation of cysteines in TGF-beta-activated kinases prevented zinc binding and recognition of ubiquitin chains[39]. We could not detect any cysteine methylation in the other C4-type zinc finger proteins present in human cytosolic ribosomes (L37, L37A, S27A, and S29) making it unlikely the observed modifications are artifacts of sample handling.

**Stoichiometry and composition of ribosomal stalk complexes.** Up to now, we focused on the benefits of bottom-up and top-down LC-MS/MS in the characterization of the composition of ribosomal particles. However, with these methods it is hard to gather any information on the stoichiometry of ribosomal proteins in the particles, an area where native MS can play an important role. The ribosomal L7/L12 stalk is a sub-complex within the large ribosomal subunit involved in the binding of several translation factors[40]. It is composed of a single copy of the

L10 protein bound by multiple dimers of the L7/L12 protein, where L7 is the N-terminally acetylated form of L12. The stoichiometry of the proteins in the ribosomal stalk depends on the number of binding sites for L7/L12 dimers on the flexible tail of L10 and can be predicted based on sequence alignment (Fig. 5a)[41]. Additionally, the stoichiometry can be determined unambiguously using a specific native MS experiment, as has been shown previously for ribosomal stalks of *Thermotoga maritima* and *Thermus thermophilus* (heptameric, L10 [L7/L12]$_6$) and *Bacillus subtilis* and *E. coli* (pentameric, L10 [L7/L12]$_4$)[42,43]. Due to the endosymbiotic origin of chloroplasts, the chloroplastic 70S ribosomes have a prokaryotic evolutionary ancestor and we set out to determine the oligomeric state of the chloroplastic stalks. Therefore, we made use of the in-source trapping activation of the recently described QE-UHMR mass spectrometer to release stalk complexes from the chloroplastic 70S ribosomes and measured their mass as being 103.4 kDa (Fig. 5b), revealing a heptameric stoichiometry[26,44]. Isolation of ions of the intact stalk complex in the quadrupole and subsequent HCD fragmentation ejected a single L12 subunit that further confirmed this assignment (Fig. 5b). We also used top-down LC-MS/MS to identify potential PTMs or processing of the stalk proteins in the Ec70S, So70S and Hs60S ribosomal particles. This revealed that unlike stalks in *E. coli*, chloroplastic ribosomes contain nearly no L7-form of the L12 protein and additionally no protein methylation could be detected (Fig. 5c). In human 60S subunits, stalks consist of the equivalent proteins P0 and dimers of P1 and P2. Our data reveal that these proteins are present as different phospho-isoforms harboring either 0, 1, or 2 phosphorylations in about equal abundance in our preparations (Fig. 5d).

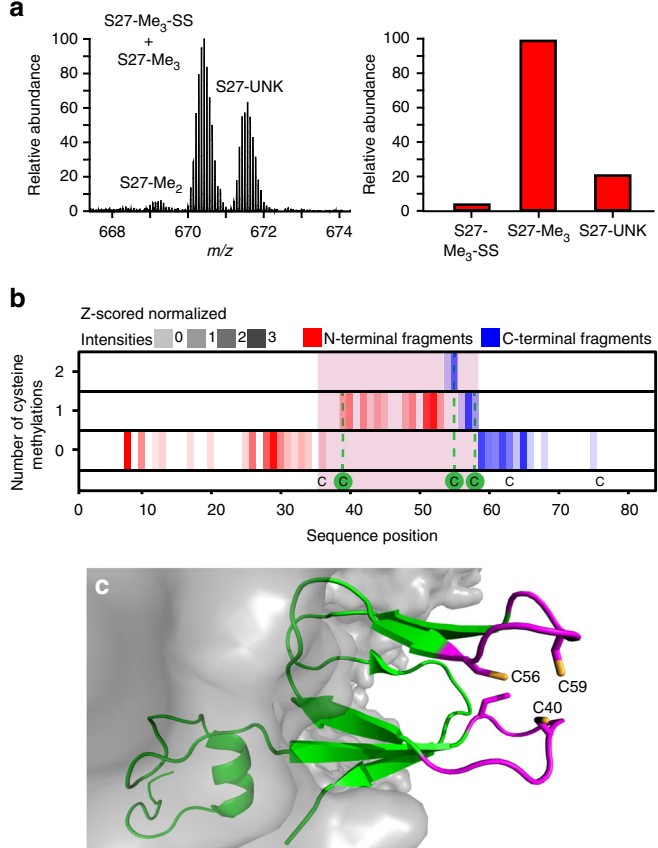

**Fig. 4** Human S27 harbors several methylated cysteines in its Zn-finger domain. **a** Intact mass spectra of S27 reveals three proteoforms with distinct masses (left). Based on the mass differences from the sequence mass of S27 these are identified as di- and tri-methylated forms of the protein (S27-Me$_2$ and S27-Me$_3$) and a third proteoform carrying an unknown modification (S27-UNK). An additional lower abundant disulfide-linked form was detected in fragmentation maps of the trimethylated form (S27-Me$_3$-SS). **b** Top-down analysis of the S27-Me$_3$ fragmentation spectra pinpoints the methylation to the cysteine residues 40, 56, and 59. Fragments are mapped along the protein sequence (vertical bars) and methylated amino acids are identified by a mass shift of 14 Da. **c** Structure of the human 40S subunit with protein S27 in green and its cysteine containing zinc finger domain in magenta (PDB 5A2Q). The zinc-finger cysteines are shown as sticks and the methylated cysteine residues are labeled

**Monitoring intact ribosomes and their association with RNA.** Above we readily showed how native MS can assist in the determination of the oligomeric state of ribosomal stalk protein complexes. However, using the recently introduced QE-UHMR mass spectrometer, we can also perform accurate high-resolution mass measurements of intact ribosomal particles, despite their high RNA content (~50%) and molecular weights in the MDa range[26,44]. For the human 40S ribosomal subunit we detected several well-resolved charge state distributions around 23,000 $m/z$ ratio (Fig. 6a). The theoretical mass of the 40S subunit based on the protein and rRNA sequences is 1,209,602 Da. The mass of the most abundant species in this spectrum, 1,215,347 ± 125 Da, deviates significantly (5.7 kDa) from this theoretical mass, more than is expected based on the high resolution of the mass spectrum and the error in our measurements. Inspection of the top-down LC-MS/MS data of the 40S subunit showed that the ribosomal protein L41 of the large 60S subunit is present at relatively high abundance (Supplementary Fig. 9, Supplementary Data 3).

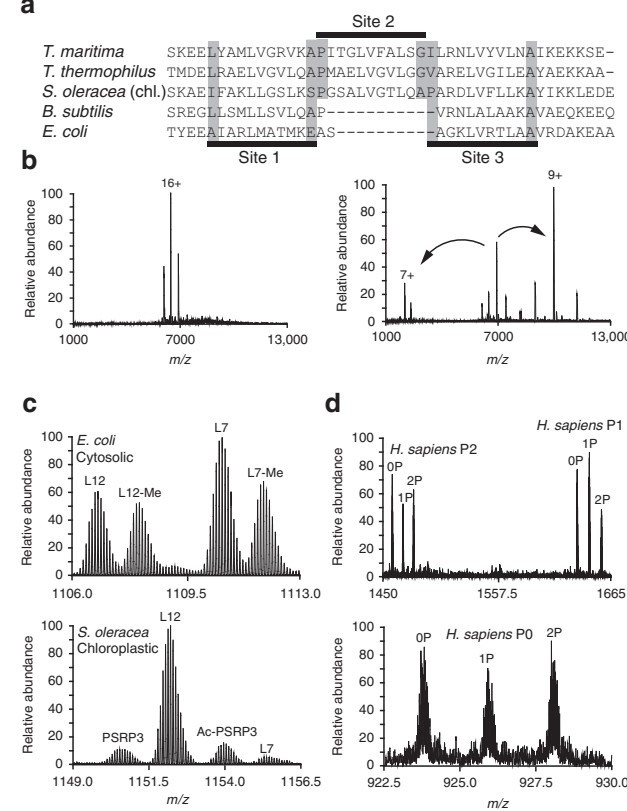

**Fig. 5** Characterization of ribosomal stalk complexes; composition, post-translational modification and stoichiometry. **a** Sequence alignment of the flexible tail region of the ribosomal protein L10 from two thermophiles (*T. maritima* and *T. thermophilus*) and two prokaryotes (*B. subtilis* and *E. coli*) with chloroplastic L10 from *S. oleracea*. The presence of three L7/L12 dimer binding sites predicts chloroplast ribosomal stalks to have a heptameric stoichiometry (L10 [L7/L12]$_6$). **b** Determination of the chloroplastic ribosome stalk stoichiometry using a pseudo-MS3 experiment. Chloroplastic 70S ribosomes are introduced into the gas phase and activated in the source region to release intact stalk complexes (top) with a mass that corresponds well with the predicted heptameric stalks. Isolation and further fragmentation of these complexes releases a single copy of the L12 protein confirming the assignment of the oligomeric state. **c** A magnificiation of a representative charge state observed in top-down LC-MS/MS analysis of bacterial (*E. coli*) and chloroplastic (*S. oleracea*) ribosomal stalk proteins shows that unlike its bacterial counterpart, chloroplastic ribosomes contain almost no L7 protein and methylation of L7 or L12 is absent. **d** Human stalk complexes in the 60S ribosome consist of the phosphoproteins P0, P1, and P2. A magnificiation of a representative charge state observed in top-down LC-MS/MS analysis reveals that all three proteins are present in their unphosphorylated (0P), singly phosphorylated (1P), and double phosphorylated forms (2P)

Furthermore, in EM reconstructions of the 40S subunit, L41 was observed to be associated to the 40S as well[37]. Addition of the mass of the L41 protein (3456.35 Da) to the theoretical mass of the 40S particle already lowered the mass deviation to ~1.7 kDa. The remaining deviation can be partly explained by the attachment of metal ions or other small molecules to the ribosomes and incomplete desolvation of the ions inside the mass spectrometer. By making use of the high resolving power of the QE-UHMR mass spectrometer, we could confidently identify other, lower abundant 40S ribosomal particles (Fig. 6a, red and purple labels) with molecular weights of 1,196,399 ± 90 Da and 1,201,827 ± 117 Da, respectively. We assign these to 40S ribosomal particles

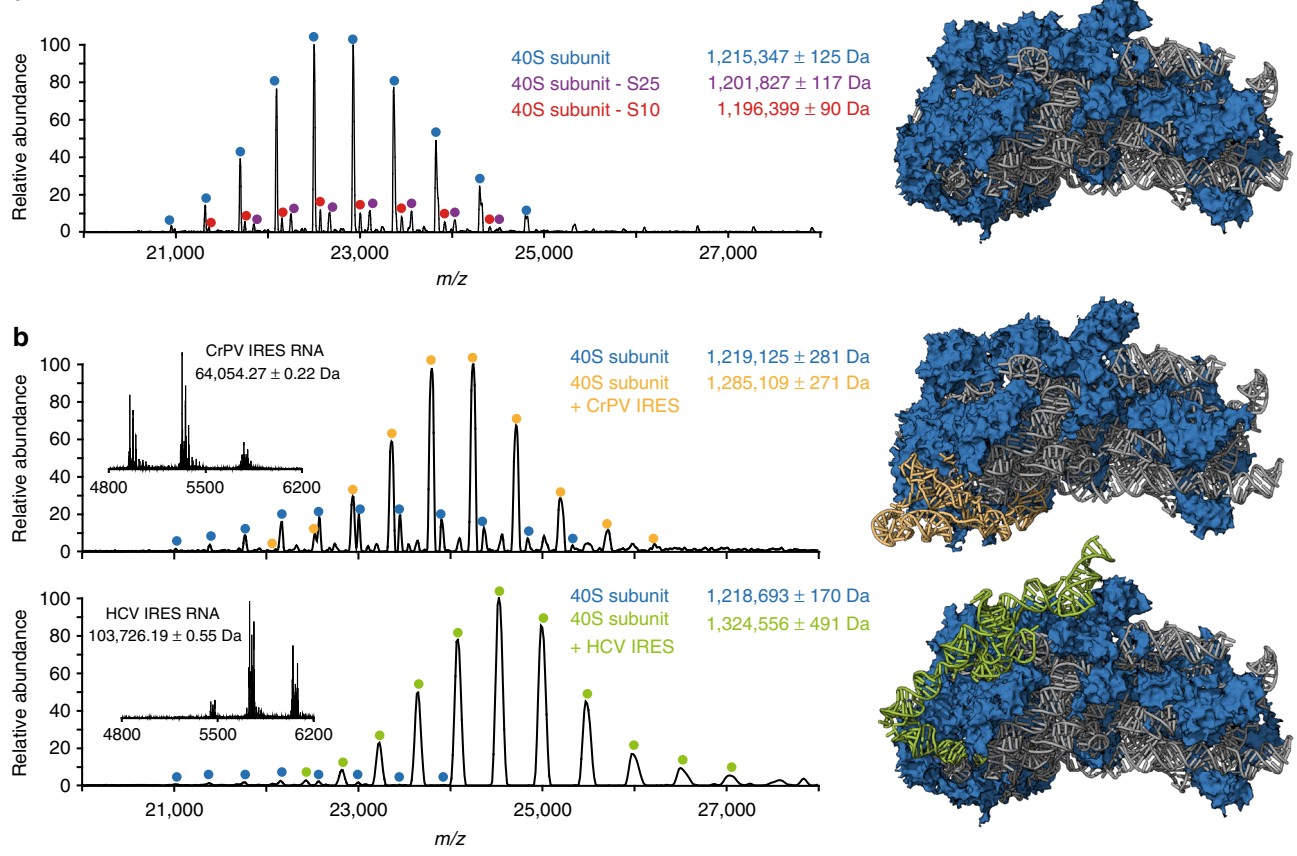

**Fig. 6** High-resolution native mass spectra of free and IRES bound human 40S subunits. **a** Native mass spectrum of human 40S subunit acquired with the recently introduced QE-UHMR mass spectrometer. The well-resolved charge states of three distinct forms of the ribosomal subunit could be detected. The most abundant fully assembled 1.2 MDa 40S particles are labeled in blue, while lower abundant particles lacking either the protein S25 or S10 are labeled in magenta and red, respectively. **b** Monitoring of the formation of a complex containing human 40S ribosomes and Internal Ribosome Entry Site (IRES) RNA fragments of Cricket Paralysis Virus (CrPV) and Hepatits C virus (HCV). Mass spectra of the RNA fragments alone (insets) provide the accurate mass of the IRES elements. This mass corresponds well with the observed increase in mass of the 40S ribosome upon binding of the RNA indicating that the particles do not undergo a significant change in composition. Structures of the free 40S ribosomes (PDB 5A2Q) and particles bound by CrPV (PDB 4V91) and HCV IRES (PDB 5A2Q) are shown, with the ribosomal proteins in blue, the rRNA in gray and the IRES elements in yellow and green, respectively[37, 47]

lacking either a copy of the ribosomal protein S10 (18.9 kDa) or S25 (13.6 kDa), respectively (Supplementary Figure 10g). Interestingly, both of these proteins were also identified as sub-stoichiometric components of human ribosomal particles in a recent selective reaction monitoring study by Shi et al.[45]. S25 was found to be mainly absent in polysomes and translation of mRNA transcripts involved in vitamin B12 pathways was influenced by the presence of this ribosomal protein. Regardless, compared to other small ribosomal subunits like Ec30S and So30S, the heterogeneity of the human 40S subunit remains rather low (Supplementary Fig. 10)[26].

Motivated by our ability to collect high-resolution native mass spectra of intact human 40S ribosomal subunits and to detect the presence or absence of individual proteins we wanted to see if we could also detect binding of functional RNA molecules to the ribosome. Therefore, we reconstituted complexes of human 40S subunits with internal ribosome entry site (IRES) RNA elements from cricket paralysis virus (CrPV) and hepatitis C virus (HCV). IRES elements are specific RNA sequences that by binding to ribosomal particles in a specific way allow for translation start while skipping certain steps of the canonical initiation process. The elements are frequently present within viral RNAs to stimulate expression of viral proteins over host cell proteins[37,46,47]. We collected native mass spectra of mixtures of free

and IRES bound 40S ribosomal particles and monitored the increase in mass upon binding of the RNA, which we determined as 66 kDa for CrPV IRES and 106 kDa for HCV IRES (Fig. 6b). Since the sequences of the IRES RNA fragments were not exactly known, we also collected high-resolution native mass spectra of the free IRES RNA structures to determine their mass. These masses (CrPV: 64,054.27 Da, HCV: 103,726.19 Da) were in good agreement with the increase in mass of the 40S subunits upon IRES binding, indicating that the ribosomal particles do not undergo a significant change in composition upon binding of the viral RNA elements. Furthermore, these data unambiguously demonstrate the 1:1 stoichiometry of IRES binding to the 40S human ribosome.

## Discussion

Structural biology has experienced remarkable developments due to technical advances in cryo-EM over the past years, culminating in being awarded the Nobel Prize in chemistry in 2017[48,49]. This sparked an increase in the number of publications featuring high-resolution structures of many important macromolecular machines[50], and especially expanded our structural knowledge about ribosomal complexes[13,14,51]. At this point, researchers have been able to describe many aspects of the translation process with significant structural detail.

Traditionally, structural biology techniques such as X-ray crystallography and NMR spectroscopy mostly required the production of artificially tagged recombinant proteins in bacterial systems. Cryo-EM, partly through its sensitivity, advanced structural biology into the field of endogenously expressed protein complexes, as is clearly exemplified by the ability of cryo-EM single particle analysis to study macromolecular assemblies purified from native sources.

Assemblies from more complex (e.g., mammalian) systems and purified from native sources display a strong increase in structural heterogeneity, both in protein and/or RNA composition and stoichiometry, but especially in the presence of chemical modifications on either the protein or nucleic acid subunits. Although some structural studies paid special attention to the importance of PTMs, the various proteoforms of the protein subunits of large protein complexes are often not detected or ignored, even though their importance is widely acknowledged[52,53]. Instances of mapping and identifying small chemical modifications in the electron density maps have been reported in literature but they required significant resolution, something which is still rarely reached using current technologies[51]. Similarly, the determination of the stoichiometry of protein and/or RNA components requires large numbers of homogenous particles images and can be hampered by compositional heterogeneity and conformational flexibility of the particle[54,55]. The techniques in the mass spectrometry toolbox can provide crucial information on all these aspects, leading to a more complete picture and understanding of macromolecular assemblies that are the subject of structural studies.

Here, we describe the in-depth characterization of four different ribosomal particles by MS using a three-tiered approach. By making use of the latest technological advances in the field of MS, most notably in top-down LC-MS/MS and native MS, our approach probes several aspects of ribosomal particles generally considered elusive to traditional structural biology methods. Above we highlighted several examples where we used the approach to gain novel insight into protein PTMs, co-occurring proteoforms and processing of ribosomal proteins, protein and RNA stoichiometry and assembly of ribosomal particles. As such, the data described serve as a proof of principle that novel, interesting aspects of large ribonucleoprotein assemblies, as shown here for diverse ribosomal particles, can be discovered using hybrid MS approaches. It lays a foundation for future studies aiming to completely characterize macromolecular complexes. For instance, the cysteine methylations of human S27 could be followed through the different stages of translation or in different organisms. Furthermore, possibly in combination with cryo-EM, more effort should be invested into fully characterizing all the proteoforms of ribosomal proteins, the exact location of these modifications, and their impact on ribosome assembly and function, as has been recently attempted for ribosomal rRNA modifications[51,56].

Technological and methodological improvements are continuously made in the field of structural biology and mass spectrometry is moving in parallel. Instruments such as the Q Exactive HF-X and the Q Exactive UHMR used here improve the depth and detail of the information that can be extracted from ribosomal particle analysis. In the future, the approach described here can be supplemented with other parts of the mass spectrometry toolbox. Mainly top-down LC-MS/MS can benefit from the use of alternative fragmentation techniques such as Electron Transfer Dissociation or Ultraviolet Photodissociation[20,57,58]. These techniques have the potential to prevent proteins with labile modifications such as glycosylation or phophorylation to escape identification and PTM localization and further improve the sequence coverage of other proteins. The presence of PTMs can greatly increase the complexity of the protein mixtures introduced in the mass spectrometer and advanced LC separations (for instance ion-exchange columns[59]) can help tackle these challenges. Additionally, advancements in data analysis software may improve the interpretation of both the raw mass spectra and large amount of information coming from all three tiers in the approach. This will make hybrid mass spectrometry approaches the ideal partner for the field of structural biology to completely unravel all details of ribosomal particles and other molecular machines[1,60]. It is evident that structure function relationships of life's cellular machineries can only be fully elucidated by the use of multiple or hybrid technologies, whereby mass spectrometry may become an indispensable pillar.

## Methods

**Names of ribosomal proteins**. Throughout the manuscript we used a short version of the Uniprot entries for ribosomal proteins. Although the nomenclature for ribosomal proteins suggested by Ban et al.[61] has been adopted in most recent structural studies, the frequent use of database searching in Uniprot in this manuscript prevented us from using this naming system.

**Purification of ribosomal particles**. E. coli 70S ribosomes were purchased from New England Biolabs. Chloroplastic 70S ribosomes were purified as previously described in detail by Bieri et al.[14]. In brief, chloroplasts were extracted from fresh leaves of spinach (S. oleracea) and lysed by gentle stirring in lysis buffer (10 mM Tris-HCl pH 7.6, 25 mM KCl, 25 mM MgCl$_2$, 2 mM DTT, 0.1 mM PMSF, 2 mM spermidine, 0.05 mM spermine, 2% (w/v) Triton X-100). The lysate was cleared by centrifugation (25,350 × $g$, 30 min, 4 °C) using a Beckman Type 45Ti rotor (Beckman-Coulter). The supernatant was loaded onto 50% (w/v) sucrose cushion and centrifuged (101,390 × $g$, 15 h, 4 °C) using a Beckman Type 45Ti rotor (Beckman-Coulter). The ribosome pellets were dissolved in monosome buffer (25 mM Tris-HCl pH 7.6, 25 mM KCl, 25 mM MgOAc$_2$, 2 mM DTT, 2 mM spermidine, 0.05 mM spermine), layered onto 10–40% (w/v) sucrose gradient and centrifuged (51,610 × $g$, 15 h, 4 °C) using a Beckman Type SW-32Ti rotor (Beckman-Coulter). The fractions containing the chloroplastic 70S ribosomes were pooled and concentrated with Amicon Ultra-4 centrifugal filter units with 100 kDa molecular weight cutoff (Merck Millipore). Aliquots of the So70S sample were flash-frozen in liquid nitrogen and shipped for MS analysis by a dry shipper (Taylor–Wharton).

Human 40S and 60S ribosomal subunits were purified similarly to previously described protocols to isolate human 80S ribosomes[37] and human 40S ribosomal subunits[62]. In brief, ~8.5 × 10$^9$ frozen HEK293-6E cells (Yves Durocher, Biotechnology Research Institute, National Research Council of Canada, 6100 Royalmount Avenue, Montreal, QC, Canada, H4P 2R2) were thawed and lysed by gentle stirring in lysis buffer (50 mM HEPES-KOH pH 7.6, 300 mM NaCl, 6 mM MgAc$_2$, 0.5% (w/v) NP-40, 5 μM E-64, 20 μM Leupeptin, 20 μM Bestatin, 5 μM Pepstatin A, 1 mM PMSF, and 2 mM DTT). The lysate was cleared by centrifugation (45,000 × $g$, 20 min, 4 °C) using a SS-34 rotor (Sorvall). The supernatant was loaded onto 60% (w/v) sucrose cushion and centrifuged (257,000 × $g$, 20 h, 4 °C) using a Beckman Type 70Ti rotor (Beckman-Coulter). The ribosome pellets were dissolved in resuspension buffer (50 mM HEPES-KOH pH 7.6, 150 mM KCl, 6 mM MgAc$_2$, 2 mM DTT), layered onto 12–48% (w/v) sucrose gradient prepared with dissociation buffer (50 mM HEPES-KOH pH 7.6, 500 mM KCl, 6 mM MgAc$_2$, 2 mM DTT) and centrifuged (78,000 × $g$, 18.5 h, 4 °C) using a Beckman Type SW-32Ti rotor (Beckman-Coulter). The bands containing the human 60S and 40S ribosomal subunits were extracted using a syringe and the fractions from several gradients were pooled and concentrated using Amicon Ultra-15 centrifugal filter units with 100 kDa molecular weight cutoff (Merck Millipore). Aliquots of the Hs40S and Hs60S ribosome samples were flash-frozen in liquid nitrogen and shipped for MS analysis by a dry shipper (Taylor–Wharton).

**Purification of IRES RNA sequences**. The CrPV IRES RNA and the HCV IRES RNA were obtained similarly as previously described by Quade et al.[37]. In brief, the IRES RNAs were produced by in vitro transcription of a linearized plasmid containing the CrPV IRES and HCV IRES sequence, respectively, followed by denaturing polyacrylamide gel electrophoresis. The IRES RNAs were extracted from the gel and the urea containing buffer was exchanged to water using Amicon Ultra-15 centrifugal filter units with 10 kDa molecular weight cutoff (Merck Millipore). Aliquots of the IRES RNA samples were flash-frozen in liquid nitrogen and shipped for MS analysis by a dry shipper (Taylor–Wharton).

**Sample preparation for bottom-up LC-MS/MS analysis**. Hs40S, Hs60S, and So70S ribosome preparations were reduced with 5 mM DTT at 56 °C for 30 min and alkylated with 15 mM iodoacetamide at room temperature for 30 min in the dark. The excess of iodoacetamide was quenched by adding 5 mM DTT. Digestion of intact proteins was performed at 37 °C with Lys-C for 4 h followed by overnight digestion with trypsin at an enzyme-to-protein-ratio of 1:100 (w/w). All proteolytic

digests were desalted, dried and dissolved in 40 µL of 0.1% FA prior to LC-MS/MS analysis.

**Bottom-up LC-MS/MS analysis.** Separation of digested protein samples was performed on an Agilent 1290 Infinity HPLC system (Agilent Technologies, Waldbronn. Germany). Samples were loaded on a 100 µm × 20 mm trap column (in-house packed with ReproSil-Pur C18-AQ, 3 µm) (Dr. Maisch GmbH, Ammerbuch-Entringen, Germany) coupled to a 50 µm × 500 mm analytical column (in-house packed with Poroshell 120 EC-C18, 2.7 µm) (Agilent Technologies, Amstelveen, The Netherlands). A 2 µL sample of peptides was used, corresponding to ~0.1 µg of material. The LC-MS/MS run time was set to 60 min with flow rate of 300 nL/min. Mobile phases A (water/0.1% formic acid) and B (80% ACN/0.1% formic acid) were used for 45 min gradient elution: 13–40% B for 35 min, and 40 to 100% B over 3 min. Samples were analyzed on a Thermo Scientific Q Exactive™ HF quadrupole-Orbitrap instrument. Nano-electrospray ionization was achieved using a coated fused silica emitter (New Objective, Cambridge, MA) biased to 2 kV. The mass spectrometer was operated in positive ion mode and the spectra were acquired in the data-dependent acquisition mode. Full MS scans were acquired with 60,000 resolution (at 200 $m/z$) and at a scan mass range of 375 to 1600 $m/z$. Automatic Gain Control (AGC) target was set to 3e6 with maximum injection time of 20 ms. Data-dependent MS/MS (dd-MS/MS) scan was acquired at 30,000 resolution (at 200 $m/z$) and with mass range of 200 to 2000 $m/z$. AGC target was set to 1e5 with maximum of injection time defined at 50 ms. 1 µ scan was acquired both for full MS and dd-MS/MS scans. Data-dependent method was set to isolation and fragmentation of the 12 most intense peaks defined in full MS scan. Parameters for isolation/fragmentation of selected ion peaks were set as follows: isolation width = 1.4 Th, HCD normalized collision energy (NCE) = 27%.

**Preparation of ribosomal proteins for top-down LC-MS/MS.** Approximately 150 µg of ribosomal proteins and ribosome associated proteins were separated from the ribosomal RNA by glacial acetic acid precipitation according to Hardy et al.[63]. Briefly, to vigorously shaken ribosomal particles, magnesium acetate was added to around 100 mM. Immediately afterwards 2 volumes of glacial acetic acid were added. This solution was left shaking at 4 °C for 60 min followed by centrifugation for 10 min at 20,000 × g. The supernatant was moved to a new vial and the pellet was washed with 66% glacial acetic acid containing 100 mM magnesium acetate. After repeating the centrifuge step the two supernatants were combined and the buffer was exchanged to buffer A (0.1% v/v formic acid in water).

**Top down LC-MS/MS analysis.** Chromatographic separation of intact protein samples was conducted on a Thermo Scientific Vanquish Horizon UHPLC system equipped with MAbPac RP 2.1 mm × 50 mm column. Around 0.5–5 µg of material was loaded on the column heated to 80 °C. LC-MS/MS runtime was set to 40 or 70 min with flow rate of 250 µL/min. Gradient elution was performed using mobile phases A (water/0.1% formic acid) and B (ACN/0.1% formic acid): 10–50% B for 30 or 60 min, and 50–80% B over additional 4 min.

All top-down MS experiments were performed on a Thermo Scientific Q Exactive HF-X instrument[27]. The instrument provides an array of new features facilitating top-down analysis of complex samples. Among them, Advanced Peak Detection (APD) algorithms, that allow for on-the-fly deconvolution of monoisotopic or average masses with improved charge detection. Along with extended charge state range (up to $z = 100+$) for calculation of optimum HCD collision energy, APD provides more efficient peak picking and fragmentation of highly charged ions of intact proteins. Improved ion optics at the front-end of the instrument results in a brighter ion source as compared with other instruments of the Q Exactive series, which provides means for less sample loads. Additionally, transmission of higher molecular weight ions (i.e., intact proteins) is improved through modified electronics and gas regime in the back-end of the instrument. LC-MS/MS data were collected with instruments set to the Intact Protein Mode.

For analysis of intact proteins two primary methods were employed, Medium-High and High-High. The former method implies switching of resolution parameter from 7500 (at 200 $m/z$) for full MS scan to 120,000 (at 200 $m/z$) for dd-MS/MS. The High-High approach defines the resolution parameter at 120,000 (at 200 $m/z$) for both full MS and dd-MS/MS. Full MS scans in both methods were acquired for the range of 400–2000 $m/z$ with AGC target set to 3e6. Maximum of injection time was defined at 250 ms with 5 µscans recorded in High-High approach and at 50 ms with 10 µscans—in Medium-High. Data-dependent strategy was focused on the three most intense proteoforms defined in full MS scan by APD algorithms. Shortly, masses corresponding to charge states of the same proteoform were deconvoluted with only the single most intense charge state selected for isolation/fragmentation in dd-MS/MS and all other charge states excluded from candidate list for user-defined exclusion time, which was optimized for each sample individually. Selected ions were isolated with 2 Th window. Collision energy applied in dd-MS/MS was normalized for the $m/z$ and charge state of selected ion with final setting of 30–35%. All the dd-MS/MS scans were recorded at the mass range of 200–2000 $m/z$ with AGC target set to 3e6 and maximum of injection time defined at 250 ms. A total of 5 µscans was recorded per scan.

**Preparation of IRES bound human 40S ribosomal particles.** Folding of IRES RNA fragments and formation of complexes with Hs40S ribosomal particles was performed according to Quade et al.[37]. Briefly, IRES RNA fragments of hepatitis C virus and cricket paralysis virus were diluted to 5 µM in folding buffer (20 mM Tris pH 7.5, 100 mM potassium acetate, 2.5 mM magnesium acetate and 250 µM spermidine) and folded by two cycles of heating to 95 °C and cooling on ice for 1 min each. Human 40S ribosomal particles were buffer exchange into binding buffer (20 mM HEPES pH 7.6, 100 mM KCl, 5 mM MgCl$_2$ and 2 mM DTT) through two cycles of concentration and dilution using a 100 kDa molecular weight cutoff centrifuge filter. Hs40S particles were mixed with a two-fold molar access of IRES RNA in binding buffer and incubated for 5 min at 37 °C and stored on ice for further use.

**Preparation of ribosomes and IRES RNA for native MS.** Hs40S, Hs40S-IRES, and So30S particles were prepared for native mass spectrometry by buffer exchanging into native MS buffer using cycles of concentration and dilution with a 10 kDa molecular weight cutoff centrifuge filter. For Hs40S particles, 1 M ammonium acetate pH 7.6 with 0.5 mM magnesium acetate was used. For So30S and Hs40S-IRES particles, 150 mM ammonium acetate pH 7.6 with 0.5 mM magnesium acetate was used. Samples were introduced into the mass spectrometer at a concentration of 100 nM after dilution with native MS buffer which contained 25 mM triethylammonium acetate pH 7.6 for the Hs40S and Hs40S-IRES particles.

Folded IRES RNA fragments were buffer exchanged to 150 mM ammonium acetate pH 7.6 using a 6 kDa Bio-Rad micro Bio-Spin column and introduced into the mass spectrometer at a concentration of 0.5 µM.

**Native MS analysis using a QE-UHMR mass spectrometer.** Samples were introduced into the Q Exactive mass spectrometer with Ultra High Mass Range detection capability (QE-UHMR) mass spectrometer with gold-coated borosilicate capillaries prepared in house[26,44]. The following mass spectrometer settings were typically used. Capillary voltage: 1350 V in positive ion mode. Collision gas: Xenon. Automatic gain control (AGC) mode: Fixed. Noise level parameter: 2. Ion transfer optics (injection flatapole, inter-flatapole lens, bent flatapole, transfer multipole and C-trap entrance lens) and voltage gradients throughout the instrument were tuned for every analyte specifically. Instrument calibration was performed using cesium iodide clusters up to 11,000 $m/z$. For Hs40S and Hs40S-IRES complexes, HCD voltage was between 250 and 300 V. For IRES-RNA fragments, HCD voltage was between 80 and 120 V. For the detection of stalk complexes of So70S ribosomes, the in-source-trapping activation voltage was optimized for maximal release and transmission of the stalks without fragmenting them further.

Spectra were viewed in Xcalibur QualBrowser software (Thermo Fisher Scientific). Masses were determined manually by minimization of the error over the charge-state envelope from different charge-state assignments. Additionally, raw native mass spectra of Hs40S, Ec30S, and So30S subunits were deconvoluted with UniDec[64] in order to obtain zero-charged mass distributions. The theoretical mass of the Hs40S particle was calculated by extracting the protein sequences from UniProtKB database (http://www.uniprot.org/uniprot/) and the rRNA sequences from NCBI.

**Data analysis for bottom-up LC-MS/MS.** Raw LC-MS/MS data were interpreted with the Byonic software suite (Protein Metrics Inc.)[65]. The following parameters were used for data searches: precursor ion mass tolerance, 10 ppm; product ion mass tolerance, 20 ppm; fixed modification: Cys carbamidomethyl; variable modification: Met oxidation. Enzymatic specificity was set to trypsin. Searches were made against UniProtKB/Swiss-Prot human and spinach proteome sequence databases. Intensity-based absolute quantification (iBAQ) values were obtained with MaxQuant software (version 1.5.6.0)[66].

**Database generation for top-down LC-MS/MS analysis.** Database searching for top-down LC-MS/MS analysis of Ec70S, Hs40S and Hs60S was performed using the *Escherichia coli* (strain K12) and Human XML format proteomes from UniProtKB.

Database searching for So70S ribosomal proteins was performed using a custom database assembled by combining sequences from *Spinacia oleracea* in UniProtKB and the BvSeq resource (http://bvseq.molgen.mpg.de). The sequences were combined in FASTA format without processing of the transit peptides.

Databases imported from XML format files in ProsightPC (Ec70S, Hs40S, and Hs60S) were treated as follows. Initiator methionine removal and N-terminal acetylation was allowed as well as other PTMs, up to 13 features or 70 kDa of features in mass per sequence. For *Spinacia oleracea* databases, no PTMs or other modifications were included in the search space.

**Data analysis for top-down LC-MS/MS.** Isotopically resolved and unresolved spectra obtained in top-down LC-MS/MS experiments of intact ribosomal proteins were deconvoluted using Xtract[67] or ReSpect algorithms (Thermo Fisher Scientific, Bremen, Germany), respectively. Automatic searches were made in Thermo Proteome Discoverer software (version 2.2.0.388) with use of ProSightPD nodes for Medium-High and High-High experimental workflows. Parameters for Medium-High method were set as follows. ReSpect parameters: precursor $m/z$ tolerance—0.2

Th; relative abundance threshold—10%; precursor mass range—5–100 kDa; precursor mass tolerance—30 ppm; charge state range—5–100. Xtract parameters: signal-to-noise (S/N) threshold—3; *m/z* range—200–2000 Th. Absolute mass search parameters: precursor mass tolerance—500 Da; fragment mass tolerance—10 ppm. In order to detect sequence variants in the So70S sample, the precursor tolerance window was initially extended up to 5 kDa. For High-High searches ReSpect parameters were not defined, instead Xtract with identical parameters was used to deconvolute spectra in both full MS and dd-MS/MS scans.

For validation of novel PTMs, HCD-MS/MS scans of the same proteoform were manually combined and fragments were assigned using in-house built fragment matching software. Intensities of assigned fragments were *z*-scored, where mean intensity was subtracted from each fragment's intensity and resulting value was divided by standard deviation of the population. The same approach was employed to characterize ribosomal proteins not detected with automated database searches. Data visualization was conducted in R extended with ggplot2 package[68]. For proteoform overview plots, monoisotopic or average masses of proteins were extracted from the precursor mass lists in automated database searches and matched with deconvoluted mass lists from MS only LC-MS experiments (with a mass tolerance window of 1 Da). Mass differences, observed mass—backbone mass (derived from UniProt sequences), were used to represent proteoforms of ribosomal proteins. Proteoform intensity was normalized on sum of proteoform intensities for each protein.

**False discovery rates in top-down LC-MS/MS database searches**. In order to estimate false discovery rates (FDR) in Ec70S, Hs40S, and Hs60S ribosomal particles, parallel searches were performed against both normal and shuffled proteome databases of *E. coli* (strain K12) and *Homo sapiens*, respectively. For So70S a normal and shuffled customized database including 77 proteoforms was used. In the cases of Ec70S and Hs40S ribosomal particles there were no protein spectral matches (PrSMs) observed when searched against reshuffled databases. For Hs60S and So70S there were 4 and 2 false-positive PrSMs detected against reshuffled databases, respectively. Further analysis showed the false discovery rate to be below 0.25% (Supplementary Fig. 11).

**Data availability**. The mass spectrometry top-down and bottom-up proteomics data have been deposited to the ProteomeXchange Consortium via the PRIDE[69] partner repository with the dataset identifier PXD008881. The native MS data generated reported in this study are available from the corresponding author upon request.

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

## Acknowledgements

M.v.d.W., S.T., K.L.F., V.F., and A.J.R.H. are funded by the large-scale proteomics facility Proteins@Work (Project 184.032.201) embedded in the Netherlands Proteomics Centre and supported by the Netherlands Organization for Scientific Research (NWO). A.J.R.H. acknowledges support by the Spinoza Prize of NWO (Project SPI.2017.028). M.v.d.W. and A.J.R.H. are also supported by a grant (12PR3303-2) from Fundamenteel Onderzoek der Materie (FOM). A.M. and A.J.R.H. acknowledge additional support through the European Union Horizon 2020 program FET-OPEN project MSmed, Project 686547. P.B. and M.I. were supported by the Swiss National Science Foundation (SNSF) and the National Center of Competence in Research (NCCR) RNA and disease programme of the Swiss National Science Foundation (SNSF). We thank Nick Quade for his support in preparation of the IRES RNAs and Joop van den Heuvel from the Helmholtz Protein Sample Production Facility for the cultivation of the HEK293-6E cells.

## Author contributions

M.v.d.W., S.T., K.L.F., E.D., and V.F. performed the experiments. M.v.d.W., S.T., and A.J.R.H. wrote the manuscript. P.B., M.I., and N.B. provided Ribosomal complexes and IRES RNA and contributed to discussions on Ribosome biology. M.v.d.W., A.M., and A.J.R.H. designed the study.

## Additional information

**Competing interests:** K.L.F., E.D., and A.M. are employees of Thermo Fisher Scientific, the manufacturer and supplier of Orbitrap-based mass spectrometers. The remaining authors declare no competing interests.

