## [Peer Review File · Nature Communications]

Reviewers' comments:

Reviewer #1 (Remarks to the Author):

The paper under review describes a mass spectrometric pipeline for characterizing ribosomal particles. The method allows determination of an impressive amount of potentially useful information concerning which ribosomal components are present in the various preparations, the presence of other specifically and non-specifically associating proteins, modifications and processing of the proteins present in these preparations, stoichiometry, as well as heterogeneity of the ribosomal particles. This type of analysis should prove of high value to researchers involved in the structural and functional characterization of entities of this type, with special value during the determination of such structures using cryo-EM techniques.

My main questions relate to the section on “Monitoring intact ribosomes and their association with RNA molecules”:

Here it is stated that the mass “...deviates significantly (6.5 kDa) from this theoretical mass, more than is expected based on the high resolution of the mass spectrum and the error in our measurements”. Based on my experience, such excess mass deviations can vary quite a bit dependent on conditions and preparation. Can you provide additional justification for this claim?

I searched for the L41 protein in the Table provided for the “bottom up” MS analysis and was unable to find it in either 60S or 40S particle components. I also could not find information on the level of this protein as ascertained by the top-down analysis. The authors should provide data to back up their claim that the “ribosomal protein L41 of the large 60S subunit is present at relatively high abundance”.

Concerning the heterogeneity observed by MS for the intact 40S ribosomal particle:

- (i) Are there any ambiguities in the assignments of – S10 and –S25 based on the observed mass differences alone?
- (ii) Is there any chance that the observed heterogeneity is produced by dissociation during particle purification, in solution, during electrospray, or in the mass analyzer? This may be worth discussing.

Reviewer #3 (Remarks to the Author):

In the paper entitled “Dissecting ribosomal particles throughout the kingdoms of life using advanced hybrid mass spectrometry methods” M. van de Waterbeemd et al. describe a combination of three mass spectrometry-based approaches (bottom-up, top-down and native mass spectrometry) for the structural analysis of ribosomal particles. Four different samples were chosen from bacterial (*E. coli*), plant (*S. oleracea*) and human origin. The main objective of this manuscript is to show the complementarity of the various approaches to obtain accurate information at the peptide, proteoform and protein assembly level (including important information such as protein processing, PTM, stoichiometry...). Although the paper presents very interesting results (and types of experiments), not all samples have been analyzed with all types of experiments and thus it is not easy to have a general view on what we can expect from combining bottom-up, top-down and native MS. The added value of top-down is clear (in particular because ribosomal proteins are very small) but the native MS part concerning the analysis of protein assemblies (without RNA) is rather poor. A single protein complex is characterized for only one sample. The manuscript would benefit from presenting more data on protein assemblies, and comparing the results obtained for the different samples (bacterial, plant, human). This would allow a real structural comparison of ribosomal particles (in term of complexity, processing, PTM, assembly...) between the different kingdoms of life (as mentioned in the title). In conclusion, this manuscript presents very interesting results obtained by a combination of state-of-the-art MS approaches, fits very well with Nature Communications but needs modifications before it can be accepted for publication.

Major points

What is the rationale behind the choice of the different samples? Why *S. oleracea* and not *A. thaliana* as a plant model, which would have simplified the database search? Is it because of the existence of cryoEM data for *S. oleracea* 70S ribosomal particle? This should be better explained in the introduction.

It seems that only top-down data of *E. coli* 70S subunit have been acquired (and neither bottom-up nor native MS data) with the aim of testing the med/high and high/high approaches. This should be more explicit in the text.

How do the distributions of theoretical MWs of ribosomal proteins compare between all samples? These distributions should be added as a Suppl. Figure.

The paper extensively describes the comparison between med/high and high/high (both with APD) but does not compare these two different approaches with or without APD. What is the added value of the APD algorithm in term of number of proteins identified for the same type of analysis (either med/high or high/high)? Is there a difference in the average measured MWs, identification rate, or number of MS/MS spectra when using APD? These data could be added as a Suppl Figure for *E. coli*.

From the data shown in the paper it is not clear to see the added value of bottom-up experiments.

How many proteins found with bottom-up were not found with top-down? How many proteins in total were found in all bottom-up experiments? What was their average size? This could be discussed in the text.

How many biological/technical replicates were performed for the bottom-up and top-down experiments?

The native MS analysis of protein assemblies (L251-275) is rather poor. A single ribosomal complex (stalk complex) is characterized with native MS experiments and for the chloroplast sample only. Why this one specifically? Was it the most abundant? Why not characterizing protein assemblies for *E. coli* and for the human sample? Only top-down data of denatured proteins are shown in Figure 5c and discussed in the text, which is rather frustrating. The authors should provide the native MS data on the other samples (*E. coli* and human). What does the Figure 5c represent exactly? Parts of MS spectra?

How is the theoretical mass of the 40S particle (protein + RNA) calculated? The authors should list the proteins used for the calculation. How is the presence of multimeric complexes taken into account? Would it be possible to fragment the intact particles to evaluate if some protein or RNA fragments are more loosely bound than others?

Minor points

L65-70: Maybe “denaturing” would be more appropriate than “unfolding”

Why do they author call their approach “medium/high” and not “Low/high”?

L138 : It should be mentioned in the text that So70S is composed of So30S and So50S.

L1/157 : « Bottom-up has no real upper limit in protein length or molecular weight ». It would be more appropriate to state that bottom-up is biased toward large proteins (small proteins are hard to detect with bottom-up) and top-down toward small ones.

L166. Since DTT is used in the ribosome purification process, the presence of reduced/oxidized proteins in top-down MS experiments could simply due to a partial reduction during sample preparation. Could the authors comment?

L193. “A sufficiently large window”. Please specify the window that was used (500 Da). A larger window could have been used.

L199. It seems from Suppl. Table 2 that it is 59 and not 57.

L396. How many cells were used?

L446. What was the amount of starting material?

L455. “Around 0.5 µg of material” This amount seems very low for the size of the column that was used (2.1 mm). Please check. The use of such column size is quite unusual, could the authors comment on their choice?

L484. Why were the bottom-up experiments performed on a Q-Exactive HF and not HF-X ?

L465 : « Improved ion optics in the front-end of the instrument result in brighter ion source ». In comparison with what other type of Orbitrap? Q-Exactive HF ? Fusion ? Fusion Lumos ?

L473 : Five or ten microscans were used for both MS and MS/MS ? Please specify in the text.

L. 496. 1000 mM ? please check.

Suppl Figure 4a: Compare with the average theoretical mass for each type of sample.

Suppl Fig 4c: Additional value of Med/high is not clear considering that it doubles the number of runs. Please comment.

Response to reviewers NCOMMS-18-03832

Reviewers' comments:

Reviewer #1 (Remarks to the Author):

The paper under review describes a mass spectrometric pipeline for characterizing ribosomal particles. The method allows determination of an impressive amount of potentially useful information concerning which ribosomal components are present in the various preparations, the presence of other specifically and non-specifically associating proteins, modifications and processing of the proteins present in these preparations, stoichiometry, as well as heterogeneity of the ribosomal particles. This type of analysis should prove of high value to researchers involved in the structural and functional characterization of entities of this type, with special value during the determination of such structures using cryo-EM techniques.

We would like to thank the reviewer for the kind remarks.

My main questions relate to the section on “Monitoring intact ribosomes and their association with RNA molecules”:

Here it is stated that the mass “...deviates significantly (6.5 kDa) from this theoretical mass, more than is expected based on the high resolution of the mass spectrum and the error in our measurements”. Based on my experience, such excess mass deviations can vary quite a bit dependent on conditions and preparation. Can you provide additional justification for this claim?

Mass deviation for protein assemblies of this molecular weight (~1 mDa) was shown to be within the 1-5 kDa range due to incomplete desolvation, especially when the non-globular shape of ribosome is taken into account (Lu, J. et al., J Am Soc Mass Spectrom, 2015, doi: doi:10.1007/s13361-015-1235-6). Additionally, we recalculated in the revised manuscript the mass of the complex taking into account the most abundant proteoforms observed in the top-down LC-full MS experiment of the Hs40S sample. This resulted in an estimated mass of 1213669 Da, which is only 1.7 kDa smaller than the centroided mass of the complex measured with native MS. It should be noted that this indicates a mass deviation of approximately 0.14%, a value similar to what has been reported previously for other assemblies in the mega-Dalton mass range, even protein-nucleic acid assemblies (van de Waterbeemd et al., Nat. Methods, 2017, doi: 10.1038/nmeth and van de Waterbeemd et al., J Am Soc Mass Spectrom, 2016, doi: 10.1007/s13361-016-1348-6).

Masses of the ribosomal proteins with their relative abundances and ssRNA molecules that were used in the calculations can be found in newly added supplementary table 3.

I searched for the L41 protein in the Table provided for the “bottom up” MS analysis and was unable to find it in either 60S or 40S particle components. I also could not find information on the level of this protein as ascertained by the top-down analysis. The authors should provide data to back up their claim that the “ribosomal protein L41 of the large 60S subunit is present at relatively high abundance”.

We thank reviewer for pointing out the missing information. Indeed, L41 protein is not identified in the bottom-up LC-MS/MS, likely because 17/25 residues in its sequence are Lys and Arg, which leads to production of very short tryptic peptides that usually are not analyzed/detected in shot-gun LC-MS/MS. The small size and high number of positively charged amino acids might also prevent efficient identification of L41 in top-down runs with automatic searches. In order to support our claims that this protein is abundantly present in the Hs40S ribosomal purification we added supplementary figure 9. The relative abundance of L41 can be found in the newly added supplementary table 3.

Concerning the heterogeneity observed by MS for the intact 40S ribosomal particle:

(i) Are there any ambiguities in the assignments of – S10 and –S25 based on the observed mass differences alone?

We understand the concerns raised by the reviewer and therefore incorporated the deconvoluted spectrum of the human 40S ribosome as well as protein matches for the observed differences for the mass variants of Hs40S subunit (new Supplemental Figure 10d, g). The low standard deviations in the mass (<150 Da) measured in figure 6 strongly reduce the ambiguity in our assignments.

(ii) Is there any chance that the observed heterogeneity is produced by dissociation during particle purification, in solution, during electrospray, or in the mass analyzer? This may be worth discussing.

This is an important point that is put forward by the reviewer, part of a broad discussion in both biochemistry and structural biology. Although we find this discussion too broad to be included in the current manuscript we do want to provide the reviewer with feedback.

It is undeniably true that the integrity of the ribosome particles may be somewhat lost during all stages between cell culture through particle storage down to mass analysis. All techniques that measure characteristics of biomolecules *in situ* or *in vitro* experience this risk, which is minimized or monitored to a varying extent. In the case of native MS analysis, dissociation upon transfer to the gas phase is not commonly detected and the field relies on ion-mobility mass spectrometry studies reporting the retaining of tertiary and quaternary structure of proteins assemblies in the gas phase (for example: Ruotolo et al., Science, 2005, DOI: 10.1126/science.1120177). Dissociation in the gas phase itself leads to characteristic asymmetric charge partitioning, distinct from in solution dissociation, which can and should be monitored and minimized during the experiment (Jurchen et al., J. Am. Chem. Soc., 2003, DOI: 10.1021/ja0211508). In the case of dissociation (or artificial formation) of assemblies in solution, a challenge inherent of nearly all techniques, not just native MS, one can often only rely on the availability of complementary information/techniques. Specifically, for the heterogeneity detected in the native mass spectra of our ribosomal particles, for both the Ec30S and the Hs40S the heterogeneity we detected was additionally shown with other, albeit more indirect, techniques (van de Waterbeemd et al., Nat. Methods, 2017, doi: 10.1038/nmeth and Shi et al., Mol Cell, 2017, DOI: <https://doi.org/10.1016/j.molcel.2017.05.021>). We expect this discussion provides useful and hopefully satisfying information to the reviewer.

Reviewer #3 (Remarks to the Author):

In the paper entitled “Dissecting ribosomal particles throughout the kingdoms of life using advanced hybrid mass spectrometry methods” M. van de Waterbeemd et al. describe a combination of three mass spectrometry-based approaches (bottom-up, top-down and native mass spectrometry) for the structural analysis of ribosomal particles. Four different samples were chosen from bacterial (*E. coli*), plant (*S. oleracea*) and human origin. The main objective of this manuscript is to show the complementarity of the various approaches to obtain accurate information at the peptide, proteoform and protein assembly level (including important information such as protein processing, PTM, stoichiometry...). Although the paper presents very interesting results (and types of experiments), not all samples have been analyzed with all types of experiments and thus it is not easy to have a general view on what we can expect from combining bottom-up, top-down and native MS. The added value of top-down is clear (in particular because ribosomal proteins are very small) but the native MS part concerning the analysis of protein assemblies (without RNA) is rather poor. A single protein complex is characterized for only one sample. The manuscript would benefit from presenting more data on protein assemblies, and comparing the results obtained for the different samples (bacterial, plant, human). This would allow a real structural comparison of ribosomal particles (in term of complexity, processing, PTM, assembly...) between the different kingdoms of life (as mentioned in the title). In conclusion, this manuscript presents very interesting results obtained by a combination of state-of-the-art MS approaches, fits very well with Nature Communications but needs modifications before it can be accepted for publication.

We thank the reviewer for his positive remarks, questions and comments. We would like to start with a response to some of the reviewer's comments on the absence of some of the experiments for some of the ribosomal particles. In our view, the main purpose of this study was to demonstrate the complementary nature of three different mass spectrometric approaches on gaining information about various aspects of large protein assemblies, in particular ribosomal particles, complementing the structural biology toolbox. Although it would have been a possibility to present data of all 3 tiers for all the ribosomal particles available to us we chose to show only specific combinations of the MS tiers and ribosomal particles. We believe that showing all tiers for all particles would lead to a manuscript of a length, unfit for non-specialized readers. Additionally, the depth of the data analysis would decrease for all tiers and may have made us miss out on some interesting details.

We selected our combinations of tiers and ribosomal particles to try and maximize the amount of data that was not previously reported. For example, native MS and bottom-up MS data of Ec30S, Ec50S and Ec70S particles was already reported by our group (van de Waterbeemd et al. Nat. Methods, 2017, doi: 10.1038/nmeth) so we complemented this with top-down LC-MS/MS data. Similarly, we decided to showcase the native MS analysis of protein(-only) assemblies using the So50S stalk since direct evidence of the stoichiometry of chloroplast stalks was not available, whereas for the *E. Coli* and human particles this was known. The stoichiometry of the cytosolic rabbit 60S ribosome stalk was previously determined (Gordiyenko et al., Mol Cell Proteomics, 2010, doi: 10.1074/mcp.M000072-MCP201) so we felt determining the stoichiometry of the Hs60S ribosome stalk (which is also of mammalian origin) was of lower interest. However, we did complement our dataset with information on the phosphorylation status of the Hs60S stalk proteins through top-down LC-MS/MS.

To further respond to the reviewer we have extended our dataset of ribosomal particles from the different kingdoms of life in the revised manuscript adding a comparison of the heterogeneity of native MS data of the small ribosome subunits (Ec30S, So30S and Hs40S) as supplementary figure 10.

Major points

What is the rationale behind the choice of the different samples? Why *S. oleracea* and not *A. thaliana* as a plant model, which would have simplified the database search? Is it because of the existence of cryoEM data for *S. oleracea* 70S ribosomal particle? This should be better explained in the introduction.

Indeed, the choice for ribosomal particles was somewhat arbitrary, reflecting which ribosome particles were already purified and studied in the lab of Nenad Ban. Difficulties in the identification of spinach proteins allowed us to claim that automatic top-down analysis still requires further development as well as to display the power of such an approach for analysis of proteins with unknown sequence and PTM variations. We thank the reviewer for pointing out that further clarifications are required. Thus we added following to the text: "Although spinach is being extensively used for structural studies, for example chloroplastic ribosomes for cryo-EM analyses are often extracted from spinach leaves^{15,35}, the spinach proteome is rather poorly described."

It seems that only top-down data of *E. coli* 70S subunit have been acquired (and neither bottom-up nor native MS data) with the aim of testing the med/high and high/high approaches. This should be more explicit in the text.

Indeed, 70S ribosome particles from *E. coli* were used primarily as a test system to optimize methods for top-down analysis, since we earlier reported on the analysis of *E. coli* ribosomes with bottom-up and native MS. This manuscript focused on the introduction of a new mass spectrometer with improved native MS analysis of ribosomal particles (van de Waterbeemd et al. *Nat. Methods*, 2017, doi: 10.1038/nmeth).

In order to add comparison of native MS of ribosomal subunits from different species we reanalyzed the previously measured Ec30S spectrum as a part of Supplementary Figure 10, wherein heterogeneity of small ribosomal subunits from human, bacteria, and spinach is compared.

How do the distributions of theoretical MWs of ribosomal proteins compare between all samples? These distributions should be added as a Suppl. Figure.

Such a comparison can already be found in Supplementary Figure 4 (a).

The paper extensively describes the comparison between med/high and high/high (both with APD) but does not compare these two different approaches with or without APD. What is the added value of the APD algorithm in term of number of proteins identified for the same type of analysis (either med/high or high/high)? Is there a difference in the average measured MWs, identification rate, or number of MS/MS spectra when using APD? These data could be added as a Suppl Figure for *E. coli*.

Unfortunately, the APD algorithm could not be deactivated on Orbitrap QE HF-X instruments hampering a direct comparison. Furthermore, comparing top-down runs on the HF-X instrument with the other instruments of the Q Exactive series would not be fair because they lack modifications in hardware and electronics that facilitate transmission and detection of intact proteins. The APD algorithm is essential for top-down experiments because it allows to detect multiple charge states of the same proteoform, select and sequence the most abundant features and exclude the others. That results in detection and sequencing of proteoforms throughout a large dynamic range of intensities. On the instrument without APD the charge states cannot be directly determined, thus neither exclusion, nor optimized collision voltage can be applied, which substantially hampers the detection of low abundant co-eluting proteoforms. The improvements on the HF-X instrument described here are of highest importance when sample complexity increases, i.e. more co-occurring proteins and proteoforms spanning large range of intensities.

From the data shown in the paper it is not clear to see the added value of bottom-up experiments. How many proteins found with bottom-up were not found with top-down? How many proteins in total were found in all bottom-up experiments? What was their average size? This could be discussed in the text.

How many biological/technical replicates were performed for the bottom-up and top-down experiments?

Bottom-up LC-MS/MS is a robust technique that provides ideal possibilities to identify proteins spanning a large dynamic range of abundances. Reliable software packages have been developed for this method, which allow for confident detection of proteins with incomplete sequence coverage, by detecting shorter size peptides. Although top-down LC-MS/MS is a very promising technique, software as well as optimal measuring parameters and conditions are still being developed and optimized. Altogether, we prefer to use bottom-up LC-MS/MS for the initial screen of sample protein content and detection of sample contaminants.

Indeed, we might have undermined the value of bottom-up experiments by focusing more on revealing exciting possibilities of the top-down LC-MS/MS. While top-down required multiple technical replicates to detect all of the proteins, bottom-up as a streamlined technique allowed us to have an explicit glimpse into sample protein composition within a single successful LC MS/MS run. In order to address this, we added the comparison of protein identifications in the bottom-up and top-down approaches as well as a comparison of protein properties for the unique hits originating the bottom-up analysis (new Supplementary Figure 6).

The native MS analysis of protein assemblies (L251-275) is rather poor. A single ribosomal complex (stalk complex) is characterized with native MS experiments and for the chloroplast sample only. Why this one specifically? Was it the most abundant? Why not characterizing protein assemblies for *E. coli* and for the human sample? Only top-down data of denaturated proteins are shown in Figure 5c and discussed in the text, which is rather frustrating. The authors should provide the native MS data on the other samples (*E. coli* and human). What does the Figure 5c represent exactly? Parts of MS spectra?

As mentioned above our choice for the chloroplast stalk is mainly a consequence of the availability of direct evidence of the stoichiometry of the *E. coli* and mammalian stalks. We would also like to note that the manuscript contains extensive native MS analysis of the Hs40S particle, alone and IRES bound. Although these complexes contain RNA and not just protein, they serve as good representatives of the native MS Tier in our approach.

We appreciate the reviewer pointing out the incomplete figure legend for figure 5C. Both figure 5C and 5D show representative charge states of intact, denatured proteins and their PTM modifications. We added this information to the legend.

How is the theoretical mass of the 40S particle (protein + RNA) calculated? The authors should list the proteins used for the calculation. How is the presence of multimeric complexes taken into account? Would it be possible to fragment the intact particles to evaluate if some protein or RNA fragments are more loosely bound than others?

In order to make the calculation of the intact protein masses clearer we added Supplementary Table 3 with masses of all the ribosomal proteins observed in the LC-full MS runs of the Hs40S sample as well as the sequence and mass of the rRNA molecule intertwined with the 40S ribosomal particle. Since the Hs40S particle does not contain proteins present at multiple copies, the presence of multimeric complexes is not taken into account.

We find the correlation between the gas-phase fragmentation efficiency and the solution phase interaction strength generally not strong enough to evaluate if some protein or RNA fragments are more loosely bound than others in our particles.

Minor points

L65-70: Maybe “denaturing” would be more appropriate than “unfolding”

We agree, and changed this throughout the text.

Why do they author call their approach “medium/high” and not “Low/high”?

To avoid confusion as Low/High is used for Ion Trap/Orbitrap experiments on the Orbitrap Fusion instruments.

L138 : It should be mentioned in the text that So70S is composed of So30S and So50S.

This missing information was added to the text:

“According to the bottom-up LC-MS/MS data, the 50 most abundant proteins in the So70S sample are either ribosomal proteins of the 30S and 50S subunit, which comprise the 70S ribosome, or translation factors (pY and RRF).”

L1/157 : « Bottom-up has no real upper limit in protein length or molecular weight ». It would be more appropriate to state that bottom-up is biased toward large proteins (small proteins are hard to detect with bottom-up) and top-down toward small ones.

We incorporated information on technique-specific biases into the following sentence: “Bottom-up LC-MS/MS provides great depth to characterize lower abundant proteins and has no bias towards the identification of larger proteins, thus with no upper limit in protein length and molecular weight, while top-down LC-MS/MS provides a more complete view of the different co-occurring proteoforms of proteins, albeit working still best within approximately 5-50 kDa mass range (Supplementary Figure 6).”

L166. Since DTT is used in the ribosome purification process, the presence of reduced/oxidized proteins in top-down MS experiments could simply due to a partial reduction during sample preparation. Could the authors comment?

While it is true that sample reduction during purification could have taken place that would not explain the formation of the observed disulfide bridges. Due to almost complete lack of information in databases about disulfide bridges in ribosomal proteins, the identified S-S links could be of potential interest. Additionally, the goal was to showcase that the top-down method has potential to successfully separate and characterize oxidized and reduced forms of proteins.

L193. "A sufficiently large window". Please specify the window that was used (500 Da). A larger window could have been used.

We used 500 Da precursor tolerance window to analyze and detect proteins with known sequences, but unknown PTMs. However, when protein sequences were not precisely known, like in case of the So70S sample, we extended the window up to 5 kDa. After correction of the protein sequences, the window would be returned to its default value of 500 Da.

The following line was added to the text in the revision: "In order to detect sequence variants in the So70S sample, the precursor tolerance window was initially extended up to 5 kDa."

L199. It seems from Suppl. Table 2 that it is 59 and not 57.

The supplementary table contains "all 57 ribosomal proteins of the chloroplastic 70S ribosome including five plastid-specific ribosomal proteins (PSRPs) as well as the translation factor pY (formerly PSRP1) and the ribosome recycling factor (RRF)", which gives in total 59 proteins.

L396. How many cells were used?

We specified the number in the text: "...approximately 8.5×10^9 of frozen HEK293-6E cells..."

L446. What was the amount of starting material?

Around 150 μg ribosomal material was used as starting material for ribosomal RNA precipitation. We now specified this number in the text.

L455. "Around 0.5 μg of material" This amount seems very low for the size of the column that was used (2.1 mm). Please check. The use of such column size is quite unusual, could the authors comment on their choice?

Indeed, 0.5 μg approximates the lower bar for the used injections of ribosomal purifications. In the revised manuscript, we added a somewhat larger range of material amount (0.5-5 μg) to the text that better reflects the used injection amounts.

The column we used was successfully employed for separation of reduced antibodies. Our choice was based on the fact that majority of the ribosomal proteins fall into the mass range of 15-60 kDa (Supplementary Figure 4a) resembling the range of masses for reduced antibody chains.

L484. Why were the bottom-up experiments performed on a Q-Exactive HF and not HF-X ?

While new features of the HF-X instrument were of significant advantage for top-down experiments, bottom-up technique was used to have a sensitive overview of the protein content and demonstrated great results already on the HF instrument with a single successful run.

L465 : « Improved ion optics in the front-end of the instrument result in brighter ion source ». In comparison with what other type of Orbitrap? Q-Exactive HF ? Fusion ? Fusion Lumos ?

We thank reviewer for pointing this out. Missing information was added to the manuscript: “Improved ion optics at the front-end of the instrument results in a brighter ion source as compared with other instruments of the Q Exactive series...”

L473 : Five or ten microscans were used for both MS and MS/MS ? Please specify in the text.

We specify the amount of microscans for full MS for two methods in the sentence: “Maximum of injection time was defined at 250 ms with 5 μ scans recorded in the High-High approach and at 50 ms with 10 μ scans – in Medium-High.”

In the last sentence of the paragraph we specify the amount of microscans for MS/MS as follows: “All the dd-MS/MS scans were recorded at the mass range of 200 to 2,000 m/z with AGC target set to 3e6 and maximum of injection time defined at 250 ms. A total of 5 μ scans was recorded per scan.”

L. 496. 1000 mM ? please check.

We used 1M ammonium acetate for native MS measurements of Hs40S. We changed 1000 mM to 1 M in the text to eliminate confusion.

Suppl Figure 4a: Compare with the average theoretical mass for each type of sample.

Supplementary Figure 4a represents distribution of **theoretical** average masses of the ribosomal proteins. To reduce possible misperception, we specified that the figure depicts “theoretical” mass distributions in the figure caption.

Suppl Fig 4c: Additional value of Med/high is not clear considering that it doubles the number of runs. Please comment.

As can be seen from supplementary figure 4c Medium/High approach leads to unique identification of 7 proteins, not taking into account some proteoforms of proteins identified in both High/High and Medium/High. To our knowledge there is no other way to identify all of the human ribosomal proteins and their abundant proteoforms but to use combinations of the methods, thus doubling the number of runs is required. Naturally, it would be possible to optimize both the chromatographic and mass spectrometer conditions to find all ribosomal proteins in a single run but since such optimized methods do not exist currently it would require a significant amount of runs to generate one and such method would mainly be applicable for a particular ribosomal protein sample.

Reviewers' Comments:

Reviewer #1 (Remarks to the Author):

The authors have satisfactorily addressed my major concerns. I therefore believe that the manuscript is suitable for publication in Nature Communications

Reviewer #3 (Remarks to the Author):

The authors addressed most of the major points in their revised manuscript. New data have been added including supplementary figures, and the text has been changed and completed to follow the reviewers' recommendations.

It is really unfortunate that it is not possible to compare top-down results with and without APD. Moreover the argument of the authors stating that "without APD charge states can not be determined" is inexact. Considering that most proteoforms are below 20 kDa (Supp Fig 4a), and that 5 microscans are summed for each spectrum, charge states could easily be resolved even without APD.

Even if, as stated by the authors in their rebuttal letter, the manuscript already contains "extensive" native MS analysis of the Hs40S particle, it would be interesting to know if beside the stalk complex, other ones (without RNA) could be measured and identified, even if these data are not shown in this manuscript? This could just be mentioned in the final version of the manuscript.

In summary this manuscript is a very nice piece of work that shows the added value of combining different proteomics approaches on the same sample.